# Fast-Embeddable Grooved Microneedles by Shear Actuation for Accurate Transdermal Drug Delivery

**DOI:** 10.3390/pharmaceutics15071966

**Published:** 2023-07-17

**Authors:** Sang-Gu Yim, Keum-Yong Seong, Akash Thamarappalli, Hyeseon Lee, Seungsoo Lee, Sanha Lee, Semin Kim, Seung-Yun Yang

**Affiliations:** 1Department of Biomaterials Science (BK21 Four Program), Life and Industry Convergence Institute, Pusan National University, Miryang 50463, Republic of Korea; sg.yim0425@gmail.com (S.-G.Y.); ky.seong0124@gmail.com (K.-Y.S.); agash.t@gmail.com (A.T.); hyeslee96@gmail.com (H.L.); sanha.lee1995@gmail.com (S.L.); 2SNVIA Co., Ltd., Hyowon Industry-Cooperation Building, Busan 46241, Republic of Korea; matisses@snvia.com (S.L.); smkim@snvia.com (S.K.)

**Keywords:** grooved microneedle, drug delivery, embedding, triamcinolone acetonide (TCA)

## Abstract

Percutaneous drug delivery using microneedles (MNs) has been extensively exploited to increase the transdermal permeability of therapeutic drugs. However, it is difficult to control the precise dosage with existing MNs and they need to be attached for a long time, so a more simple and scalable method is required for accurate transdermal drug delivery. In this study, we developed grooved MNs that can be embedded into the skin by mechanical fracture following simple shear actuation. Grooved MNs are prepared from hyaluronic acid (HA), which is a highly biocompatible and biodegradable biopolymer. By adjusting the aspect ratio (length:diameter) of the MN and the position of the groove, the MN tip inserted into the skin can be easily broken by shear force. In addition, it was demonstrated that it is possible to deliver the desired amount of triamcinolone acetonide (TCA) for alopecia areata by controlling the position of the groove structure and the concentration of TCA loaded in the MN. It was also confirmed that the tip of the TCA MN can be accurately delivered into the skin with a high probability (98% or more) by fabricating an easy-to-operate applicator to provide adequate shear force. The grooved MN platform has proven to be able to load the desired amount of a drug and deliver it at the correct dose.

## 1. Introduction

Over the past two decades, there has been extensive research on microneedles (MNs) as a viable option for transdermal drug delivery in response to the growing demand for painless and patient-friendly drug delivery methods [1,2,3,4,5,6,7,8,9]. With lengths ranging from 50 to 2000 microns and widths ranging from 10 to 300 microns, MNs can penetrate the outermost layer of dead skin, known as the stratum corneum, to effectively reach viable cells and interstitial fluid in the epidermis layer for optimal drug delivery [8,9]. In terms of drug delivery mechanisms, MNs can be categorized into five types: solid, hollow, coated, dissolving, or hydrogel-forming MNs [1,2,3,4,10,11,12]. Recently, cryogenic MNs have been developed for the transdermal delivery of living cells [13]. Each type offers unique advantages and limitations that require careful consideration while selecting an appropriate MN-based drug delivery strategy. Solid MNs deliver drugs into the skin by creating microchannels [1,9]. Disadvantages of this method include the risk of infection and the challenge of quantitatively delivering the drug. In contrast, hollow MNs possess holes that can inject a drug solution into the skin, similar to a syringe, and allow for rapid drug delivery of a larger volume [14,15]. The fabrication process for hollow MNs is often complex, and the holes in the MNs may become obstructed by tissues. Solid MNs can be coated with a drug solution to create coated MNs, which necessitates coating equipment and can result in higher fabrication costs [3]. Dissolving MNs are a viable alternative and are commonly used by researchers due to their simple and cost-effective fabrication method, as they are prepared using water-soluble polymers containing drugs [1,9,16]. Dissolving MNs dissolve completely in a certain time after being inserted into the skin without biological hazardous waste. However, the mechanical strength of the MNs can be weakened by the concentration of the drug, as the drug is contained within the MNs [17]. Additionally, the amount of delivered drug may vary based on the depth of its insertion into the skin [18]. In situations where a large amount of bodily fluid is secreted or when MN is attached for an extended period, the MN may dissolve up to its base, resulting in excessive drug delivery.

While the MN-mediated transdermal drug delivery system has been proven effective, ensuring precise drug delivery into the skin remains a critical obstacle to its acceptance among the pharmaceutical industries for commercialization [19,20]. To address this issue, various types of multi-layered MNs were developed. These dissolving MNs were characterized by the ability to separate or selectively dissolve the drug-loaded tip from the shaft after insertion into the skin [21,22,23].

Chen et al. reported the fabrication of a multi-layered MN array that achieved complete and sustained delivery of vaccines to the skin. The MN consisted of a water-soluble chitosan tip and a non-dissolvable polylactide acid (PLA) supporting shaft. The scalability of the process was limited by a labor-intensive fabrication process involving multiple molding steps, including partial filling of water-soluble and drug-loaded polymer layers using centrifugation, followed by integration with the PLA supporting layer [24]. Lahiji et al. employed a micro-pillar-based injector as an alternative approach and accomplished complete drug delivery to the skin by enabling the insertion of dissolving MNs loaded with drugs after physically detaching them from the substrate [25]. The micropillar-assisted injection system allowed for the implantation of powder-carrying MNs into the skin for high-dose bioactive agents such as insulin. However, this approach required invasive implantation of the device. In a separate investigation, an insertion-responsive MN (IRMN) was developed to achieve accurate drug delivery [26]. This IRMN comprised two separable layers at the drug-loaded tip and base, which were prepared using a two-step tip filling and base integration procedure. Upon insertion of the device into the skin, the mechanical stress exerted on the double-layered IRMNs caused them to crack and separate between the drug-loaded tips and the base. The drug-containing tips in the IRMN array were quickly separated upon insertion into the skin for 10 s and were fully retained at the insertion site after removing the baseplate. Nevertheless, the multiple fabrication steps that involve baseplate alignment require further refinement to guarantee complete needle deposition. More recently, rapidly separable MNs with bubble structures have been reported for the long-term release of contraceptive hormones such as levonorgestrel (LNG) [27]. The bubble MN loaded with LNG was fabricated through a two-step process. Firstly, a biodegradable polymer was used to fill a reusable mold, followed by the loading of a thin backing layer soluble in bodily fluid. After insertion into the skin, this bubble MN rapidly separated and achieved a sustained release of LNG, maintaining a high therapeutic level in the plasma for one month, as evidenced by animal studies. Yet, the restricted drug loading capacity in the separable tip presents a challenge to implementing this system in a human study that requires a high dose. In order to surpass the limitations of current MN platforms and enhance MN-based drug delivery systems in the future, a simpler and more scalable approach is required to achieve precise transdermal drug delivery.

In this study, we developed a dissolving grooved MN embeddable into the skin by mechanical fracture following simple shear actuation. Grooved MNs were structurally stable and rapidly embedded, overcoming the limitations of the long-time skin attachment of conventional MNs. In addition, the desired amount of drug was precisely delivered by fracture at the defined groove neck position. The grooved MNs were fabricated by a one-step molding process with a reusable polydimethylsiloxane (PDMS) mold. The incorporation of a groove structure into the bullet-shaped MN shaft enabled easy breaking of the MN tip at the groove neck upon application of shear force (Figure 1). We prepared the grooved MNs using hyaluronic acid (HA), a highly biocompatible and biodegradable biopolymer and investigated the impact of structural factors such as MN length, aspect ratio (length:diameter), and groove neck position on shear-induced fracture and embedding rate. To enhance drug delivery efficacy, we used a custom-designed applicator for insertion. Furthermore, we loaded triamcinolone acetonide (TCA) in the grooved HA MN array and demonstrated precise dose control by adjusting drug loading amounts. Our findings lay the foundation for the potential application of this grooved MN platform in the treatment of hair loss.

## 2. Materials and Methods

### 2.1. Materials

Sodium hyaluronate (HA, 100 kDa) was purchased from SNVIA Co., Ltd. (Busan, Republic of Korea). Polydimethylsiloxane (PDMS, Sylgard 184) was purchased from Dow Corning (Midland, MI, USA). Triamcinolone acetonide (TCA) was purchased from Sigma-Aldrich (St. Louis, MO, USA). HPLC-grade ethanol and methanol were purchased from Thermo Fisher Scientific Inc. (Waltham, MA, USA).

### 2.2. Finite Element Analysis

To confirm whether the grooved MNs could be embedded into the skin by external forces, the distribution of stress on the moment was simulated by finite element analysis (COMSOL Multiphysics, COMSOL Inc., Stockholm, Sweden, v4.3a) using the yield strength of a HA film. Additionally, simulations were performed to determine the radius and position of the groove on the MNs. The HA film specimens were prepared by applying 10 wt% of HA to a glass plate with a doctor blade and dried at 40 °C overnight with a thickness of 100 μm. The HA film in the form of a dog bone (ASTM D1708) with a length of 22 mm and a width of 5 mm was measured using a mechanical tester (AND 2000, AND, Daegu, Republic of Korea) for yield strength. The HA film was stretched until it was broken at a constant rate of 0.1 mm/min (*n* = 5).

### 2.3. Fabrication of TCA-Loaded Grooved MN Arrays

To fabricate grooved MNs with different aspect ratios (MN height to MN width), master MN pins were milled using a computer numerically controlled (CNC) machine. A positive metal master mold was fabricated by arranging an 8 × 8 array in an area of 1.1 × 1.1 cm^2^ using the master MN pins. The negative PDMS mold was fabricated with the help of the positive metal master mold [10,28]. To fabricate TCA-loaded MN patches, aqueous TCA-HA dispersion was used for the molding process. In total, 100 mg TCA powder was thoroughly dispersed in 5 mL distilled water (DI water) by sonication for 30 min. Additionally, 0.5 g of HA was added to 5 mL of the aqueous TCA solution and mixed for 12 h using a roller mixer to prepare the aqueous TCA-HA dispersion. After dropping 0.3 mL TCA-HA dispersion on the PDMS mold, the TCA-HA dispersion was filled in the cavities of the mold while reducing pressure in a vacuum chamber and dried overnight at 40 °C with a relative humidity of 55%. The shape of the fabricated metal master MN pins and TCA-loaded MNs was confirmed by a DINO microscope camera and an optical microscope (TS100, Nikon, Tokyo, Japan). The dimension of the MN structure was measured using ImageJ software. TCA-unloaded HA MN, a control group, was prepared in the same process except for TCA.

### 2.4. Evaluation of Mechanical Properties

The compression mode of a mechanical tester was used to analyze the mechanical properties of TCA-loaded MNs according to their aspect ratios [28,29]. After cutting from the 8 × 8 MN array patch, a single MN was attached to the pin mount stub using cyanoacrylate glue and then fixed to the lower grip of the mechanical tester. Another pin mount stub with nothing attached was fixed to the upper grip of the mechanical tester and moved at a constant speed of 0.1 mm/min in the axial direction. The compression force was measured against the movement of the probe as a function of displacement. HA MNs without TCA were used as a control.

### 2.5. Measurement of Shear Fracture Force

The shear fracture force of TCA-loaded MNs, according to their aspect ratios, was measured by the shear mode of a mechanical tester. After cutting an MN in an 8 × 8 array, each MN was attached to the pin mount stub using cyanoacrylate glue and then fixed to the lower grip of the mechanical tester such that the MN tip structure was perpendicular to the axial direction. A glass slide was fixed to the upper grip of the mechanical tester and moved at a constant speed of 0.1 mm/min in the axial direction. The compression force was measured against the motion of the probe as a function of displacement. HA MNs without TCA were used as a control.

### 2.6. Quantitative Evaluation of TCA Loaded in MN

To confirm the amount of TCA in the MN tip according to its aspect ratio, the tip was cut at the position of the groove using a razor blade. The cut MN tips were placed in a 1.5 mL tube, and 1 mL of 50% ethanol aqueous solution was added to the tube to dissolve the tips completely. The resulting solution was filtered using a commercial syringe filter (pore size: 0.2 μm). The amount of TCA in the MN tips was analyzed by high-performance liquid chromatography (HPLC; e2695, Waters, Milford, MA, USA). HPLC was performed using the following equipment: 2498 UV/visible detector with analytical flowcell and SunFire C18 column (100 Å, 5 μm, 4.6 × 250 mm) [30]. Methanol: DI water (60:40, *v*/*v*) was used as the mobile phase, and the flow rate was 1 mL/min. The detection wavelength was set at 254 nm. The TCA solution (20 μL) was injected into the instrument, and the analysis temperature was set at 30 °C. Standard TCA solutions in the range of 7.8–500 μg/mL were analyzed to quantify the amount of TCA. Additionally, we attempted to confirm the specifications of the MN patches fabricated according to the TCA concentration. Briefly, 4, 5, 10, and 20 mg/mL of TCA aqueous dispersion were prepared, and 10 wt% of HA was added to the TCA aqueous dispersion. The TCA-loaded MN patches were fabricated by casting 0.3 mL of TCA-HA dispersion on a PDMS mold. The amount of TCA in the MN tips was analyzed by the same method as above.

### 2.7. Ex Vivo Drug Release Test Using Franz Diffusion Cells

Static diffusion experiments were conducted using Franz cells to investigate the ex vivo skin permeation kinetics of TCA released from the embedded MN tips [30]. The TCA-loaded MNs (including approximately 150 μg of TCA/patch) were applied to the excised rat skin (approximately 1 mm thick). The MN array punctured the 3 × 3 cm^2^ excised rat skin and positioned it between the donor and receptor chambers in the Franz diffusion cell. The receptor chamber was filled with fresh phosphate-buffered saline buffer (PBS, 20 mL, pH 7.4) and maintained at 37 °C. The samples (1 mL) were removed from the receptor chamber at pre-determined time points (0, 0.5, 1, 2, 4, 8, 12, 24, 48, and 72 h), and fresh PBS buffer (1 mL) was refilled into the receptor chamber. The amount of permeated TCA was quantified using the HPLC assay described in Section 2.6.

### 2.8. Animals Experimental Protocols and In Vivo Application of TCA-Loaded MNs Using a Customized Applicator

The protocol for the animal experiment used in this study was reviewed and approved based on ethical procedures and scientific care by the Pusan National University-Institutional Animal Care and Use Committee (PNU-IACUC; PNU-2015-0951). The penetration rate of grooved TCA-loaded MNs was evaluated using SD rats (eight-week-old males, 250–300 g, Samtako Bio Korea, Osan, Republic of Korea). SD rats were anesthetized using 3% isoflurane, a commonly used inhalation anesthetic. A hair clipper first epilated the hair on the back of the anesthetized SD rat in a 4 × 8 cm^2^ area (approximately), and the second epilation was performed using a depilatory cream. The shaved area was wiped with 70% ethanol for the application of MNs. The TCA-loaded MN patch was attached to a hydrocolloid patch and further connected to the customized applicator. The TCA-loaded MN patch was attached to a hydrocolloid patch and further connected to the spring-loaded, hand-held applicator. The insertion speed of the patch was estimated to be 0.37 m/s. This estimation was obtained by employing Equation (1) related to the free vibration of a single-degree-of-freedom system [31].
(1)$$v_{max}=\sqrt{\frac{k}{m}\;}\times A,$$
where $v_{max}$ is the maximum speed of the moving part in the applicator. k, m, and A are the constant, mass, and travel distance of spring-loaded in the applicator, respectively.

After securing the site to apply MNs by pulling the skin at the shaved area with a hand, the MN was inserted vertically for 10 s using an applicator, during which shear force was manually applied to the applicator. The intrusion rate in the skin was evaluated by counting the number of embedded MN tips after taking a picture of the application area.

## 3. Results and Discussion

### 3.1. Simulation to Determine the Structure of the Grooved MN

The simulation was performed to select the optimal MN structure in which fracture may occur due to the groove structure. The simulation was conducted under two conditions: (i) the size of the MN is the same, while the position and radius of the grooves are different, and (ii) the position and radius of the grooves are the same, while the diameter of the MN is different. Specifically, in the first condition, the radius of the groove was 50, 75, and 100 μm, and the height from the groove to the end of the MN cylinder was 150, 250, and 350 μm (Table 1). Our results confirmed that fracture occurred in the MN structures when the forces applied to the groove were higher than the yield strength of HA (7.84 ± 1.10 MPa), except for the cases in which the height was 250 or 350 μm and the radius of the groove was 50 μm (Figure 2). In the second condition, the MN diameter was altered such that the aspect ratio was 3.5:1, 4.0:1, and 4.5:1 (MN length to MN diameter), and the simulation was carried out. In all conditions, it was confirmed that fracture occurred under the maximum stress of 12.78, 12.34, and 11.43 MPa, respectively, which was higher than the yield strength of HA (Figure 3). Through this, it was verified that only the tip of the MN could be embedded by adjusting the position, groove radius, or diameter of the MN.

### 3.2. Fabrication of Grooved Microneedle Patches

To deliver the drug contained in the MN tip into the skin quantitatively, it is advantageous that the MN tip is cut at the boundary point where the skin is inserted. Approximately 70% of the total length of the MN is inserted into the skin due to the elasticity of the skin; therefore, we introduced a groove structure at a position of approximately 70% of the total height of the MN based on simulation results (Figure 4). Additionally, the overall shape of the MN was fabricated into a bullet-shaped structure known to be stable for skin insertion. The master metal pin introduced the groove structure, was milled through metal cutting, and was fabricated by varying the aspect ratio of the MN. The master metal mold was fabricated using a master metal pin, and the negative PDMS mold was fabricated using metal molds. A TCA-loaded MN patch with an 8 × 8 MNs array per 1.1 × 1.1 cm^2^ area was fabricated by casting a HA solution loaded with TCA (20 mg/mL), a model drug, on the fabricated negative PDMS mold. Since TCA has low solubility (20 μg/mL) in water, the TCA aqueous solution and the prepared MN patches are shown in an opaque white color (Figure 4). The aspect ratios of the fabricated metal MNs were 3.2, 3.6, and 4 (tip height = average 800 μm; base diameter = 249 ± 5, 227 ± 4, 204 ± 5 μm, respectively; groove diameter = 187 ± 3, 186 ± 3, 171 ± 1 μm, respectively) (Table 2). The TCA-loaded MN fabricated according to the same aspect ratios (3.2 and 3.6) reduced the volume by 10–20% compared to the metal MN (tip height = average 800 μm; base diameter = 214 ± 10, 192 ± 3 μm, respectively; groove diameter = 161 ± 7, 154 ± 6 μm, respectively) (Table 2). The TCA-loaded MN made from a mold with an aspect ratio of 4 suffered deformation due to shrinkage upon drying; hence, the dimension could not be measured. Since the surface energy of the PDMS mold is low (22–25 mJ m^−2^) [32], wetting was inefficient, and shrinkage occurred during the drying process. As a result, the dried TCA-loaded MN came out of the PDMS mold without any breakage despite the groove structure. Therefore, subsequent analyses were performed using TCA-loaded MNs fabricated with an aspect ratio of 3.2 and 3.6.

### 3.3. Mechanical Properties

According to the aspect ratio of the MN (AR 3.2 and AR 3.6), the compression fracture test and shear fracture test of the TCA-unloaded HA MN (HA MN) and the TCA-loaded MN (TCA MN) were performed using a customized mechanical testing machine. For the compression fracture test, a single MN affixed on the bottom mount was gently compressed using a flat metal plate (as shown in Figure 5a). Figure 5b shows the force–displacement curve obtained for HA MNs and TCA-loaded MNs with different aspect ratios. Yield force is the point at which the end of the MN tip begins to bend or deform. The yield force of the TCA-loaded MN tended to be lower than that of the HA MN (Figure 5c), possibly due to the plasticizing effect of TCA in HA polymer chains. The compression fracture force for each MN was quantitatively measured over 0.3 N/needle (Figure 5d), which provided enough force to penetrate the skin without breaking [28]. Next, a single MN fixed horizontally on the bottom mount was compressed using a glass slide for the shear fracture test (as shown in Figure 6a). Figure 6b shows the force–displacement curves obtained for TCA-loaded MNs and HA MNs with different aspect ratios. The shear fracture force of TCA-loaded MNs tended to be lower than that of HA MNs (Figure 6c). Shear failure occurred when a force of 0.12 N or more was applied to each MN. Therefore, the groove structure introduced into MNs causes no breaking when inserted vertically into the skin and can be easily broken when an external force such as shear force is applied after insertion into the skin.

### 3.4. Quantification of TCA Loaded in the MNs

The specific quantity of the drug present in the grooved MN tip can be delivered accurately into the skin because the MN tip is broken at the groove. TCA, a synthetic corticosteroid drug, is widely used as an anti-inflammatory agent applied to the skin or mucous membranes in the oral cavity in topical formulations such as creams, gels, and lotions [33,34]. The amount of TCA in the MN tip was quantified using HPLC analysis after cutting the MN tip at the position of the groove and dissolving it in 50% ethanol. The TCA loading amount of MNs prepared using 20 mg/mL of TCA dispersion was 148.6 ± 10.3 and 128.4 ± 6.9 μg/patch for the aspect ratios 3.2 and 3.6, respectively (Figure 7a). When MNs with an aspect ratio of 3.2 were used, and the concentration of the TCA dispersion was changed to 4, 5, 10, and 20 mg/mL, it was able to control the amount of TCA in the MNs from 19.5 ± 3.7 to 144.8 ± 10.7 μg/patch (Figure 7b). In clinical applications, TCA is used as a treatment for alopecia areata [35]. Alopecia areata is characterized by sudden hair loss in a circular pattern. It can occur mainly on the scalp (and rarely affects the beard, eyebrows, and eyelashes) and causes hair loss, which may increase as the affected area increases [36]. Although the cause of alopecia areata is not clear, it is known as a systemic autoimmune disease in which T cell lymphocytes do not recognize hair as part of the body and attack the hair follicles or inhibit the growth of hair. For the treatment of alopecia areata, TCA is delivered directly through intradermal injection into the scalp; 5 mg/mL of TCA dispersion is injected directly from 0.1 to 3 mL/cm^2^ using a 30 G needle for several months at intervals of 4 to 6 weeks [37,38]. Therefore, since the amount of TCA in the MN tip satisfies the amount of TCA used to treat alopecia areata, TCA-loaded MN can be used for the treatment of alopecia areata.

### 3.5. Ex Vivo TCA Release Test Using Franz Diffusion Cells

For application to the treatment of alopecia areata, the release behavior of TCA in the MN tips (AR 3.2) was analyzed using a Franz diffusion cell (Figure 8). The amount of TCA in the MN tips was approximately 150 μg/patch, corresponding to one application/3 days. After applying the TCA-loaded MNs to rat skin, it was transferred to a Franz diffusion cell. The TCA-permeated solution was sampled at the pre-determined time and quantitatively analyzed using HPLC. It was confirmed that the TCA in the MN tips was released over a long time, more than 80%, for 72 h in the rat skin (Figure 8). Recently, a method using solid MN-based micro-needling has been reported to deliver TCA into the scalp [39]. After applying a dermaroller containing a 1.5 mm long needle to the affected area (five times in the horizontal and vertical directions), a 10 mg/mL concentration of TCA dispersion was applied. However, dermarolling requires professionals for the procedure, and bleeding or inflammatory reactions may occur due to repeated treatment using the dermaroller. Additionally, it is difficult to deliver TCA to the scalp quantitatively. Therefore, TCA can be delivered using MNs that easily penetrate the skin and can be applied by the patient himself after receiving a prescription at a hospital. Furthermore, it is confirmed that the drug can be delivered quantitatively over a long period of time. To evaluate the efficacy of TCA-loaded MNs, anti-inflammatory pharmacological tests on cell or animal models will be performed in future work.

### 3.6. In Vivo Application of TCA-Loaded MNs

The previously studied dissolving MNs should be applied to the skin and attached until the MN tip is completely dissolved. If the dissolving MN is not well inserted into the skin, the MN tip does not dissolve, and if the backing layer containing the drug dissolves, it is difficult to deliver the drug quantitatively. A customized applicator was used to accurately and vertically insert the prepared MN patches into the skin and apply adequate shear force, as shown in Figure 9a. A cartridge for fixation of the TCA-loaded MN patch was set at the bottom of the applicator, pressing the upper button vertically inserted the MN patch into the skin. MN patches with an aspect ratio of 3.2 were used. SD rats were anesthetized to remove the hair on the dorsal area of the body for the application of MN patches in vivo. After securing a space for using the applicator by pulling the skin taut, the MN patch was inserted by applying the applicator to the skin, and a force was applied in the shear direction to induce the fracture of the MN tip (Figure 9b). By checking the skin on the dorsal area of the rat, it was confirmed that the fractured MN tips were inserted into the skin (Figure 9c). Therefore, the MNs with groove structure could easily embed the MN tips into the skin through an applicator. This can be utilized as a patchless system capable of quantitatively delivering drugs without attaching the patch for a long time. Since the grooved MN array can be fabricated by a one-step molding process, this structure-based embeddable MN would be advantageous for mass production compared to multi-layered separable MNs prepared by multiple fabrication steps. However, the application method involving manual breaking via shear actuation needs to be improved to facilitate the utilization of the grooved MN in clinical settings. To achieve accurate intradermal drug delivery and a cost-effective MN manufacturing system, a coated MN system that offers selective multiple coating on the MN tip would be an alternative [40].

## 4. Conclusions

In this study, we developed a new embeddable MN system by introducing a groove structure into the bullet-shaped dissolving MN. The grooved MN array enabled rapid and accurate implantation of MN tips into the skin by following a simple ‘press-and-slide’ step assisted by a spring-loaded hand-held applicator. The loaded amount of TCA, a treatment for alopecia areata, was quantitatively evaluated by varying the position of the groove and the TCA concentration in the MN-forming solution. These grooved MNs showed that the desired amount of drug could be accurately delivered by controlling the position of the groove structure or the concentration of TCA loaded on the MN. TCA-loaded MN tips were more than 95% embedded in the skin by applying shear force after application. Furthermore, TCA was released over a long time by passive diffusion through the MN tips embedded in rat skin. Grooved MNs are expected to increase patient convenience and provide an accurate drug dosage. Moreover, they can be used for the application of several drugs due to their high-speed drug delivery system and long-term drug release capability.

## Figures and Tables

**Figure 1 pharmaceutics-15-01966-f001:**
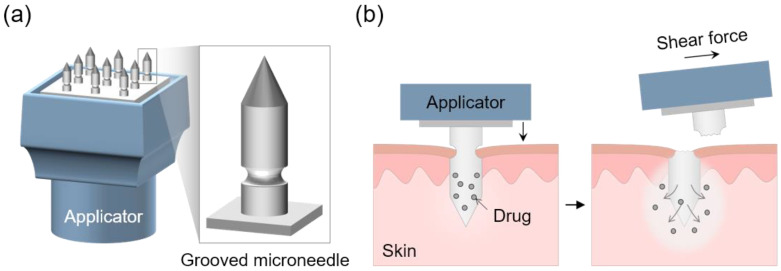
Schematic illustration showing (**a**) the grooved MN array installed on the applicator and (**b**) the implantation mechanism of the grooved MN by shear actuation after insertion into the skin using the MN applicator.

**Figure 2 pharmaceutics-15-01966-f002:**
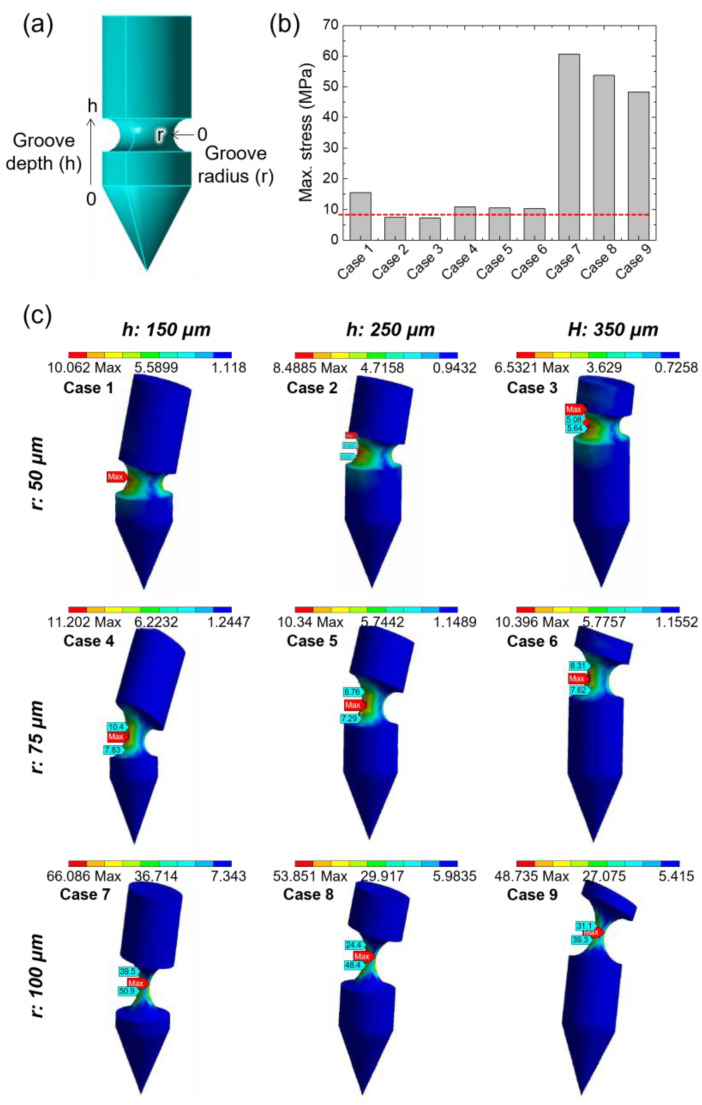
Finite element analysis of stress distribution in the bullet-shaped MN having different groove specifications under shear stress. (**a**) Representative modeling image of the grooved MN showing both variables. The total length and base diameter of the MN is 900 μm and 250 μm, respectively. (**b**) Maximum stress exerted on the bullet-shaped MN having different grooves, as shown in Table 1. The red line represents the yield strength of the sodium hyaluronate (HA) film. (**c**) Stress distribution in the bullet-shaped MN with different grooves under shear stress.

**Figure 3 pharmaceutics-15-01966-f003:**
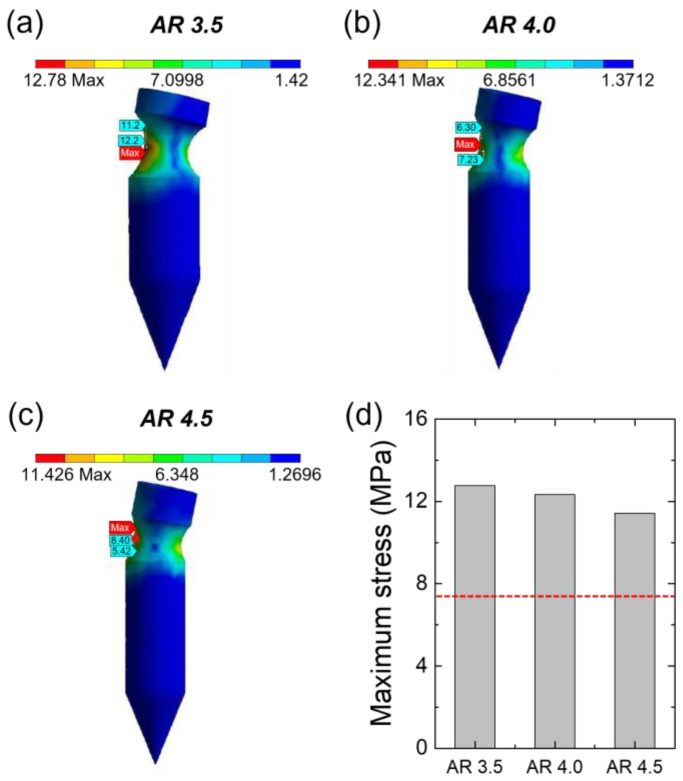
(**a**–**c**) Finite element analysis of stress distribution showing the effect of different aspect ratios (height to radius ratio, ARs) on the bullet-shaped grooved MN; (**a**) AR 3.5, (**b**) AR 4.0, and (**c**) AR 4.5. (**d**) Maximum stress exerted on the bullet-shaped MN with different ARs. The red line represents the yield strength of the HA film.

**Figure 4 pharmaceutics-15-01966-f004:**
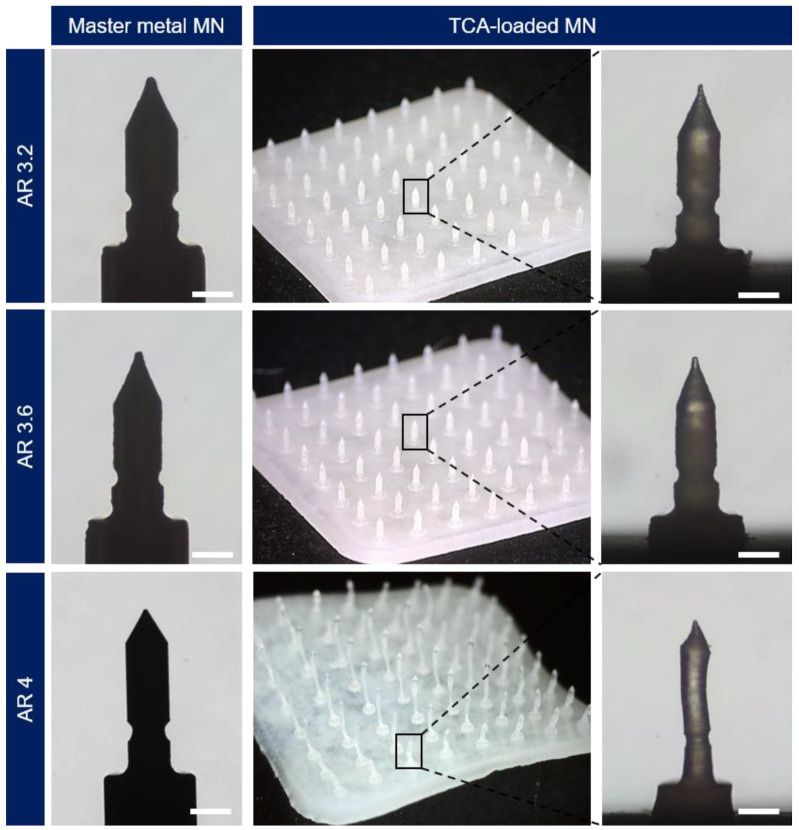
Photographic images showing micromachined metal MNs with different ARs used as a master mold and TCA-loaded grooved HA MNs prepared using a PDMS negative mold (scale bar: 200 μm).

**Figure 5 pharmaceutics-15-01966-f005:**
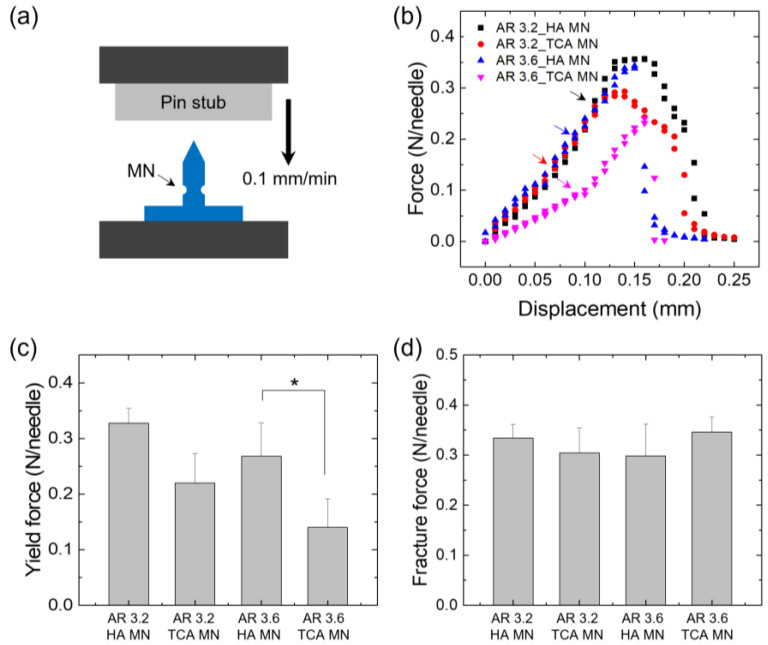
(**a**) Schematic illustration showing the compression test of grooved MNs. (**b**) Representative force–displacement profiles obtained for grooved HA MNs and TCA-loaded grooved HA MNs (TCA MN) with two different aspect ratios (3.2 and 3.6) in a compression mode. (**c**) Yield force and (**d**) fracture force of HA MNs and TCA MNs with different aspect ratios obtained in the compression tests (*n* = 5). The asterisk (*) indicates statistical significance with *p* < 0.05.

**Figure 6 pharmaceutics-15-01966-f006:**
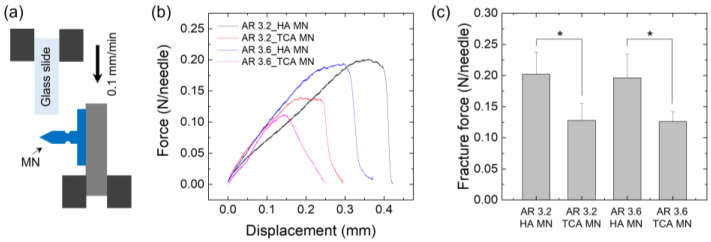
(**a**) Schematic illustration showing the shear fracture test of grooved MNs. (**b**) Representative force–displacement profiles obtained for HA MNs and TCA MNs with two different aspect ratios (3.2 and 3.6) in a shear fracture mode. (**c**) Fracture force of HA MNs and TCA MNs with different aspect ratios in the shear fracture tests (*n* = 5). The asterisk (*) indicates statistical significance with *p* < 0.05.

**Figure 7 pharmaceutics-15-01966-f007:**
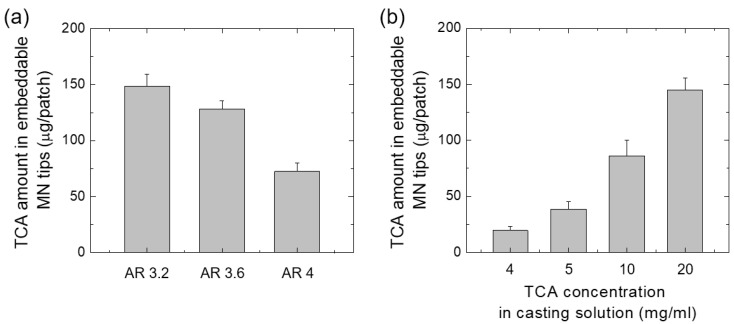
(**a**) TCA loading amounts in the embeddable tips of a TCA MN patch (8 × 8 MNs) with different ARs. (**b**) TCA loading amounts in the embeddable tips of a TCA MN patch with AR 3.2 prepared using different casting (MN-forming) solutions containing different TCA concentrations.

**Figure 8 pharmaceutics-15-01966-f008:**
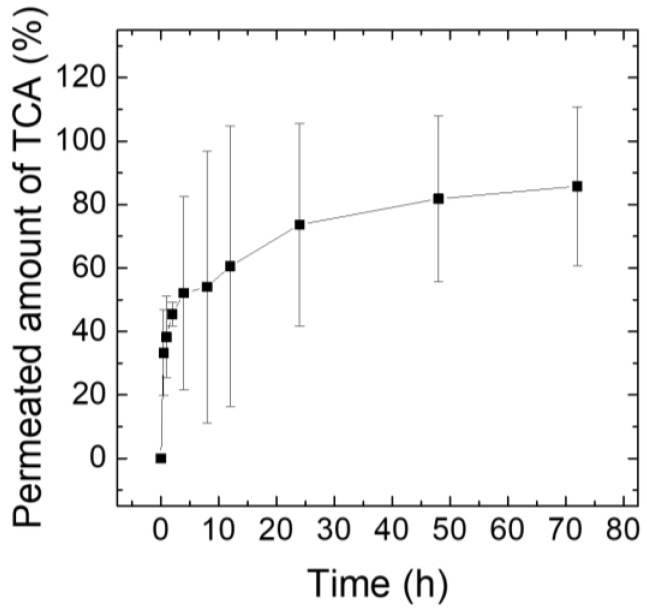
Ex vivo permeation profiles of TCA across rat skin following application of a TCA MN array with AR 3.2 containing approximately 150 μg of TCA in embeddable MN tips (*n* = 3).

**Figure 9 pharmaceutics-15-01966-f009:**
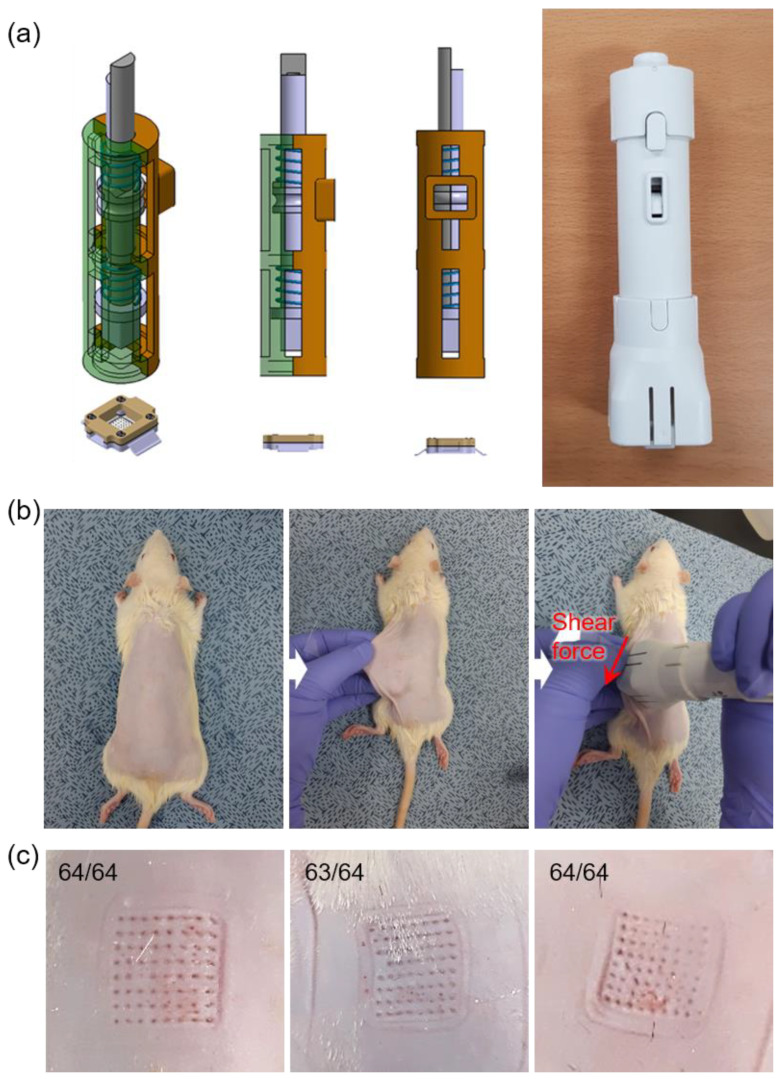
(**a**) A customized MN applicator used in this study. (**b**) Photos showing the application procedure of the MN array (8 × 8 MNs in 1.1 × 1.1 cm^2^ area) installed on the applicator to the shaved dorsal area of the rat. After insertion of the MN array into the skin by using the applicator, the MNs were implanted by manually applying shear force. (**c**) Photos showing the embedded MNs as punch marks on rat skin after removal of the applicator.

**Table 1 pharmaceutics-15-01966-t001:** Dimensions of the grooved MNs.

	Grooved MNs								
	Case 1	Case 2	Case 3	Case 4	Case 5	Case 6	Case 7	Case 8	Case 9
Groove radius (r, μm)	50	50	50	75	75	75	100	100	100
Groove depth (h, μm)	150	250	350	150	250	350	150	250	350

**Table 2 pharmaceutics-15-01966-t002:** Dimensions of metal MNs and TCA-loaded MNs shown in Figure 4.

Aspect Ratio	Metal MN			TCA-Loaded MN		
	3.2	3.6	4	3.2	3.6	4
Tip height (μm)	812 ± 9	820 ± 5	791 ± 5	804 ± 5	801 ± 11	N.A.*
Base diameter (μm)	249 ± 5	227 ± 4	204 ± 5	183 ± 6	192 ± 3	N.A.*
Groove diameter (μm)	187 ± 3	186 ± 3	171 ± 1	138 ± 3	154 ± 6	N.A.*

* N.A., Not available.

## Data Availability

The data that support the findings of this study are available from the corresponding authors upon reasonable request.

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
