# Peer review of "Fast-Embeddable Grooved Microneedles by Shear Actuation for Accurate Transdermal Drug Delivery"

_pharmaceutics, 2023, doi:10.3390/pharmaceutics15071966_

Round 1

Reviewer 1 Report

In this manuscript, Yim et al. describe the design and characterization of grooved microneedles (MNs), made of hyaluronic acid (HA), that can be embedded into the skin by applying shear force for precise transdermal drug delivery. The length, diameter, and position of the groove on the MNs was adjusted to ensure successful MN insertion. Moreover, grooved MNs loaded with triamcinolone acetonide demonstrated accurate dose control by adjusting drug loading amounts. The paper is interesting for broader scientific community, presents of a modern solution for delivering drugs to the skin. Overall, it is well written; however, there are some areas in the manuscript, which should be addressed to increase the clarity and strengthen the article:

1. Line 50-51: “However, the mechanical strength of the MNs can be weakened by the concentration of the drug, as the drug is contained within the MNs.” It depends on several factors such as the polymer and the drug, so I would suggest to state that the mechanical strength of MNs can be affected by the concentration of the drug.

Please find the following references:

https://doi.org/10.1039/D2BM01143C

https://doi.org/10.1016/j.jmbbm.2021.104384

2. Line 58-61: “In response, various types of separable MNs were later developed as second-generation MNs. These MNs were characterized by the ability to separate or selectively dissolve the drug-loaded tip from the base after insertion into the skin.” I would like to suggest considering the following recent papers to provide additional support and ensure a more comprehensive coverage of this topic. In these papers, the drug was concentrated at the tips rather than dispersed in the base layer to provide more efficient drug delivery:

https://doi.org/10.1016/j.ejpb.2020.10.002

https://doi.org/10.3390/pharmaceutics15020526

https://doi.org/10.1016/j.bioadv.2022.212729

3. In section 2.2: Please provide more details on the fabrication process including the amounts and volumes of the polymer and drug.

4. In section 2.6: the authors mentioned that the TCA-loaded MNs (including approximately 30 μg of TCA/patch) were applied to the excised rat skin. Are the MNs included only 30 μg of TCA/patch? It should be approximately 7 mg.

5. In section 2.7: Please provide more information about the customized applicator? The force applied, the insertion rate (speed) etc.

6. Line 271-272: I suggest to use “MN array” instead of “a single MN” for the sake of clarity. Additionally, I think that you meant Figure. 5a instead of Figure. 2a.

7. Figure 5b: The figure should be improved because it is difficult to see the differences among the different samples and the AR values are 3.2 and 3.6. Furthermore, I suggest to highlight the yield force points by arrows for example.

8. Line 274-276: The authors mentioned that “The yield force of the TCA-loaded MN tended to be lower than that of the HA MN, which is believed to be a consequence of the plasticizing effect of TCA in HA polymeric chains”. Did you perform DSC analysis to confirm that?

9. Line 277-278: “compression fracture force for each MN was quantitatively measured over 0.3 N/needle, which provided enough force to penetrate the skin without breaking (Figure. 5c).” Please provide references.

10. Figure 6b: The figure should be improved (e.g. line instead of dots).

11. Line 300-303: Please provide references.

12. Line 304-306: “The TCA loading amount of MNs prepared using 20 mg/mL of TCA dispersion was 148.6 ± 10.3 and 128.4 ± 6.9 μg/patch for the aspect ratios 3.2 and 3.6, respectively”. As mentioned above, it is not clear what is the actual/obtained drug loading, e.g. you mentioned in section 2.6 that the TCA-loaded MNs (including approximately 30 μg of TCA/patch). Please clarify.

13. Line 309-318: Please provide references!

14. I appreciate the thoroughness of the manuscript and the authors' efforts to draw meaningful conclusions. However, I believe that the discussion could be more developed.

Reviewer 2 Report

This article reported a microneedle system (“fast-embeddable grooved microneedles”) read interesting. The concept was actually good, and this manuscript fell within the scope of Pharmaceutics. However, the reviewer had some concerns regarding the current version, and this work was somehow incomplete. A substantial revision must be conducted before a second decision.

Detailed comments:

A. In Line 36-37, the categories of MNs should be updated. There were gel MNs and cryo-MNs nowadays. And a new reference should be cited.

B. In the last paragraph of Introduction, the authors introduced the technical details of the grooved MNs. However, herein the advantages of this kind of MNs over conventional ones must be mentioned.

C. An individual section about the materials used for this study must be provided.

D. According to Table 1, the “height” of grooved MNs was 150~350 μm. Actually, it was not the tip height of grooved MNs (~800 μm, which was shown in Table 2). This might lead to unnecessary misunderstanding. Please consider to change the expression.

E. The standard deviation of Figure 5 and 6 associated with mechanical tests was a bit large. Please comment on the reproducibility of the method.

F. In addition to the mechanical tests, the authors were advised to exam the penetration effects of grooved MNs on rat/mice/pig skins, with the help of Trypan blue staining.

G. Although the authors tested the in vivo application performance, the data was to preliminary to demonstrate the superiorities of grooved MNs. The reviewer suggested to study at least one of the following on animal models: bioavailability, skin drug retention, skin irritation and biosafety.

H. Before the Conclusion Section, it was recommended to discuss the industrialization and clinical translation aspects of the designed system.

I. Since the authors employed TCA as the model drug, could they consider to perform anti-inflammatory pharmacological tests on cell or animal models

Round 2

Reviewer 2 Report

I have no further questions. But I still suggest the authors to consider to perform some pharmacological tests in the future.